# Validation of Lead-DBS β-Oscillation Localization with Directional Electrodes

**DOI:** 10.3390/bioengineering10080898

**Published:** 2023-07-28

**Authors:** Colette Boëx, Abdullah Al Awadhi, Rémi Tyrand, Marco V. Corniola, Astrid Kibleur, Vanessa Fleury, Pierre R. Burkhard, Shahan Momjian

**Affiliations:** 1Department of Neurosurgery, University Hospitals of Geneva, CH-1205 Geneva, Switzerlandshahan.momjian@hcuge.ch (S.M.); 2Faculty of Medicine, University of Geneva, 1206 Geneva, Switzerlandpierre.burkhard@unige.ch (P.R.B.); 3Department of Neurosurgery, Pontchaillou Hospitals, CEDEX 9, F-35033 Rennes, France; 4Centre Hospitalier Universitaire Caen Normandie, F-14000 Caen, France

**Keywords:** local field potentials, lead-DBS, subthalamic nucleus, Parkinson’s disease

## Abstract

In deep brain stimulation (DBS) studies in patients with Parkinson’s disease, the Lead-DBS toolbox allows the reconstruction of the location of β-oscillations in the subthalamic nucleus (STN) using Vercise Cartesia directional electrodes (Boston Scientific). The objective was to compare these probabilistic locations with those of intraoperative monopolar β-oscillations computed from local field potentials (0.5–3 kHz) recorded by using shielded single wires and an extracranial shielded reference electrode. For each electrode contact, power spectral densities of the β-band (13–31 Hz) were compared with those of all eight electrode contacts on the directional electrodes. The DBS Intrinsic Template AtLas (DISTAL), electrophysiological, and DBS target atlases of the Lead-DBS toolbox were applied to the reconstructed electrodes from preoperative MRI and postoperative CT. Thirty-six electrodes (20 patients: 7 females, 13 males; both STN electrodes for 16 of 20 patients; one single STN electrode for 4 of 20 patients) were analyzed. Stimulation sites both dorsal and/or lateral to the sensorimotor STN were the most efficient. In 33 out of 36 electrodes, at least one contact was measured with stronger β-oscillations, including 23 electrodes running through or touching the ventral subpart of the β-oscillations’ probabilistic volume, while 10 did not touch it but were adjacent to this volume; in 3 out of 36 electrodes, no contact was found with β-oscillations and all 3 were distant from this volume. Monopolar local field potentials confirmed the ventral subpart of the probabilistic β-oscillations.

## 1. Introduction

Deep brain stimulation of the subthalamic nucleus (STN-DBS) has been a treatment for motor symptoms in patients with Parkinson’s disease (PD) for more than 30 years. Increased β-oscillatory activity was highlighted as a pathological activity within the basal ganglia of these patients, which decreased during voluntary movements or after dopaminergic medication [1,2]. Indeed, dopaminergic medication, as well as STN stimulation, induces changes in β-oscillations, which were found to be associated with improvements in motor symptoms [3]. In addition, β-oscillations in the STN were shown to be principally located in the sensorimotor subdivision of the dorsolateral STN [4].

The Lead-DBS imaging reconstruction toolbox has recently been made available to the scientific community (www.lead-dbs.org (accessed on 21 January 2022)) [5]. It allows postoperative localization of the implanted cerebral DBS electrodes with different atlases, in particular the DISTAL atlas [6] that defines population average-based functional subdivisions of the STN, an atlas with population-defined DBS targets [7], and an electrophysiological atlas for β- and α-oscillations [8]. Nonetheless, few studies have attempted to compare these localization tools with real intraoperative electrophysiology. The concordance between microelectrode recordings and the DISTAL atlas in comparison to other anatomical atlases of the Lead-DBS toolbox has been verified [9,10].

A new directional electrode, the Vercise Cartesia electrode, has also recently been made available (Boston Scientific, Marlborough, MA, USA) for DBS in PD. This technology allows stimulation of segmented contacts in order to customize the stimulation for each patient [11]. The definition of the DBS target in PD is currently still a matter of debate. Indeed, in addition to the stimulation of the STN per se, the stimulation of afferent or efferent tracts of the STN has also been proposed.

The primary objective of this study was to compare the location of β-oscillations proposed by the electrophysiological atlas implemented in the Lead-DBS software (version 2.3.1) with the location of real intraoperative β-oscillations measured by using the Vercise Cartesia directional electrodes from the recordings of local field potentials (LFPs) of each of their eight contacts. The second objective was to analyze the efficacy of stimulated anatomical structures in improving motor symptoms.

## 2. Materials and Methods

### 2.1. Patients and Clinical Assessment

This retrospective study was approved by our local Ethics Committee (“Commission cantonale d’éthique de la recherche de Genève”, CCER, Geneva CE N° 2020-02010). The ethical guidelines of the Declaration of Helsinki were applied. All patients gave their approval, either for general research or for the reuse of their data linked to DBS, by signing the dedicated forms.

The targeting method and the DBS surgical procedure were the same as those described earlier [12]. During surgery, patients were awake while microelectrode recordings were taken and intraoperative macrostimulation was performed for determining the threshold of corticospinal tract excitation. The final trajectory was chosen when STN cells could be electrophysiologically identified and also depended on the clinical observation of corticospinal tract stimulation on the contralateral face and upper limb, with a stimulation threshold of ≥2.5 mA.

Intraoperative measurements of the LFPs were performed at the time of the study in 20 patients with PD who agreed to participate (7 females, 13 males; median age 61.5 years, percentiles: 25th, 51.2 years; 75th, 70.0 years). The contacts of the left hemisphere electrodes were numbered from E1 (the most ventral contact) to E8 (the most dorsal contact) and those of the right hemisphere from E9 to E16. The intermediate contacts were segmented into groups of three.

The motor outcome was assessed 1 year post-surgery. Improvements in scores on the Movement Disorders Society-Unified Parkinson’s Disease Rating Scale (MDS-UPDRS) [13] part III were assessed under OFF-drug/ON-stimulation conditions 1 year post-surgery (overnight withdrawal of dopaminergic treatment, routine evaluation in four conditions of medication and stimulation) versus an OFF-drug condition 1 month before surgery. The changes in motor scores were assessed as percentages of the preoperative MDS-UPDRS motor score. The lateralized scores for the contralateral limbs included the items 3.4, 3.5, 3.6, 3.7, and 3.8 for bradykinesia; for rigidity, the item 3.3; and for tremor, the items 3.15, 3.16, and 3.17.

The stimulated contacts at the evaluation of motor outcome 1 year post-surgery were collected for analysis of the site of stimulation with its motor efficacy. The stimulated contacts were those identified during the clinical management of these patients in their first year of DBS.

### 2.2. β-Oscillation Measurements

The β-oscillations were computed from LFPs (0.5–3 kHz) recorded on each electrode contact, whatever its location (no contact selection). LFPs were recorded in a monopolar configuration, with the extracranial skin flap as a reference, using one of the sterilized and shielded wires of the Neurostar system (Neurostar, Tübingen, Germany) connected to the skin flap through an alligator clip. The wires connecting the electrode contacts were also shielded up to their connection with the Mephisto amplifier (Neurostar, Tubingen, Germany). Thanks to this shielding and to the use of an extracranial reference, high quality β-oscillation measurements were obtained for every contact, eliminating the intrinsic contamination proper to the referencing of bipolar measurements or to intracerebral referencing. LFP recordings lasted at least 12 s.

Postoperatively, for each contact, the spectrograms were computed by using fast Fourier transforms on non-overlapping windows, achieving power spectral densities (PSD) with 1 Hz and 1 s resolution (MATLAB, MathWorks, Natick, MA, USA). Epochs containing artefacts were identified by visual inspection and rejected. For each contact, the mean power of the β-oscillations was computed by averaging the successive PSDs of the spectrogram of the (13–31 Hz) frequency band [12].

The differences in LFP power due to differences in the surface areas of the Vercise Cartesia contacts were not integrated into the analyses. Indeed, integrating these differences would have implied an increase in the power of non-directional contacts, which, due to larger surfaces, have a lower impedance in comparison with the directional contacts, as has been verified by others [14]. To evaluate this difference, we measured a 21 Hz sinusoidal wave on electrode contacts bathed in a sodium chloride solution (0.9%). The LFPs were, as expected, less powerful on non-directional than on directional contacts (15%; Appendix A). We refer to the non-integration of that difference in the Discussion.

### 2.3. Postoperative Image Reconstruction

The DBS electrode trajectories were postoperatively reconstructed by using the Lead-DBS MATLAB toolbox (version 2.3.1; https://www.lead-dbs.org/ (accessed on 21 January 2022); [5]). The default pipeline was applied; advanced normalization tools (https://stnava.github.io/ANTs/ (accessed on 21 January 2022)) allowed the co-registration of brain images, combining the preoperative 3D T1-weighted (repetition time (TR) = 1930 ms, echo time (TE) = 2.36 ms, slice thickness 1 mm; referred to as “anat_t1” in the Lead-DBS toolbox), T2-weighted (TR = 2400 ms, TE = 225 ms, slice thickness 1 mm; referred to as “anat_t2” in the Lead-DBS toolbox), and FLAIR (TR = 5000 ms, TE = 386 ms, slice thickness 1 mm; referred to as “anat_t2star” in the Lead-DBS toolbox) MRIs (Skyra 3.0 T scanner, Siemens Medical Systems, Erlangen, Germany) with the postoperative CT scan performed the day after surgery (slice thickness between 0.6 and 1.25 mm, pixel spacing of 0.453/0.453; Somatom Definition Flash, Siemens Medical Systems). After co-registration, Lead-DBS performed an automatic correction for brain shifts [15]. As applied in an earlier study [9], based on the preoperative volumes, the symmetric image normalization diffeomorphic mapping method [16] was used to compute multispectral normalization to the ICBM 2009b nonlinear asymmetric space (Montreal Neurological Institute, MNI; [17]). The unified segmentation method [18] of the Statistical Parametric Mapping software (SPM12; https://fil.ion.ucl.ac.uk/spm/ (accessed on 21 January 2022); [19]) was applied when the previous approach was unsuccessful. The PaCER method was applied to pre-construct the DBS electrodes (without manual correction [20]). The orientation of the electrode contacts was not corrected with the DiODe technique [21].

The reconstructed images were segmented with the DISTAL atlas [6], which brought out all the relevant subcortical structures, including the STN subdivisions. The anatomical structures in relation to each electrode contact were determined by superimposing an atlas that brought out subcortical structures, including the STN subdivisions (i.e., sensorimotor, associative, or limbic), the substantia nigra, the thalamus, the nucleus reticulatus polaris (surrounding and enfolding the thalamus), and the zona incerta (ZI). This atlas was made from manual segmentations of a high-resolution brain template series (MNI, 152 template series), to which an atlas of histology and an atlas of structural connectivity were co-registered.

The probabilistic volumes containing β-oscillations by the Lead-DBS toolbox [8] were applied (electrophysiological atlas of the STN activity, β-oscillations). At the time of developing this toolbox, the probabilistic location of β-oscillations was built from bipolar recordings of an externalized Medtronic electrode (quadripolar 3389 lead; 2 mm distance between centers of two adjacent contacts; Medtronic, MN, USA). Power values (7–35 Hz) were collected over a group of 51 patients. The “power values were mapped onto subcortical anatomy of the brain in (MNI) space”. “Each datapoint was mapped to the Euclidean midpoint between the coordinates representing the two electrode contacts from which the signal was recorded” [8]. At the time of developing this toolbox, the probabilistic location of the STN-DBS target was built from a group of 39 patients with PD by using the active contacts 1 year post-surgery.

### 2.4. Statistical Analyses

For each electrode, the contacts that presented significantly stronger β-oscillations were determined by comparing the PSDs (with 1 Hz and 1 s resolution) of the β-oscillations (7–35 Hz) of each contact with the PSDs of all eight electrode contacts, for an even more competitive comparison. For this purpose, two-tailed Student’s t-tests were applied (MATLAB).

The differences in β-oscillations across contacts were expressed as percentages of the mean PSDs of all eight electrode contacts.

Analysis of variance (ANOVA) on ranks was used to compare improvements in the MDS-UPDRS motor scores between groups of patients from the location of the stimulated contacts (Kruskal–Wallis, SigmaPlot; in the situation where the equal variance test failed *p* < 0.05).

## 3. Results

### 3.1. β-Oscillations

Intraoperative measurements of the LFPs were performed for 36 electrodes (both STNs for 16 patients, a single STN for 4 patients due to intraoperative time constraints). An example of the LFPs, recorded over all eight contacts of an electrode, is illustrated in Figure 1 (which shows the right electrode of patient P4; note the high quality of LFPs obtained thanks to the shielding wires). The stronger β-oscillations measured on contact E9, bottom trace, are visible, even before performing statistics. This contact was found with stronger β-oscillations compared with those computed over all contacts of the electrode (130% stronger than the mean of β-oscillations measured on all eight contacts, *p* < 1.10^−12^).

Figure 2 illustrates an example of the Lead-DBS electrode location with the probabilistic locations of β-oscillations (burgundy volume; [8]), the STN (DISTAL atlas, sensorimotor: orange, limbic: yellow; associative: blue; [6]), and the STN-DBS target (red volume; [7]) in patient P16 in a posterior view. Note that the probabilistic locations of β-oscillations presented a banana-like shape in the dorsolateral sensorimotor STN with a wider ventral volume. Here, the most ventral contacts of the electrode of both STNs, E1 and E9 were found with significantly stronger β-oscillations than those computed over all the contacts of the electrode (two-tailed Student’s t-tests: +72%, *p <* 1.10^−4^, and +45%*, p* < 1.10^−3^, respectively). In addition, the most dorsal contact of the left electrode, contact E8, also had stronger β-oscillations than those computed over all contacts of the electrode, but with a lower significance than that of the most ventral contact (*p* < 1.10^−2^ vs. *p* < 1.10^−4^). This contact was located in the Campus Forelii (fields of Forel). The sites of stimulation are indicated with red contacts; here, two contacts are dorsal to the site of measured and probabilistic β-oscillations in the sensorimotor subregion of the STN, and one contact is ventral to the site of measured β-oscillations in the Campus Forelii.

Figure 3 shows the location of β-oscillations determined by Lead-DBS and the measured contacts with β-oscillations for the whole group of patients. For all patients (P), the yellow boxes indicate electrode contacts, as represented in the bottom row and mimicking the conformation of the electrodes (1 to 8 on left electrodes, L; 9 to 16 on right electrodes, R) found in or touching the probabilistic volume with β-oscillations by the Lead-DBS. The asterisks indicate contacts measured with significantly stronger β-oscillations than those computed over all contacts of the electrode (* *p* < 0.01, ** *p* < 0.001, *** *p* < 0.0001). Appendix A illustrates the same probabilistic locations and measured contacts with β-oscillations, together with their statistics, for anterior and posterior views (bottom left of figures). 

In 23 of the 36 measured electrodes, in 17 of the 20 patients, at least one contact was found in or touching the ventral subpart of the probabilistic volume with β-oscillations (Figure 3, at least one yellow-filled cell per column; Figure 4a). In 20 of these 23 electrodes, at least one of the contacts found in or touching the ventral subpart of the probabilistic volume with β-oscillations was also measured with significantly stronger β-oscillations than those computed over all contacts of the electrode (Figure 3, yellow-filled cells and black stars). In one of these 23 electrodes, the stronger β-oscillations were measured on a contact touching the dorsal subpart of the probabilistic volume with β-oscillations (patient P1, left electrode). In 2 of these 23 cases, the stronger β-oscillations were measured on a contact either dorsal or just ventral to the probabilistic volume (patients P6 and P8, left electrodes). 

For 10 of the 36 measured electrodes, in 10 of the 20 patients, no contact was located in or touching the ventral subpart of the probabilistic volume with β-oscillations; however, at least one contact was measured with significantly stronger β-oscillations than those computed over all contacts of the electrode (Figure 3, no yellow-filled cells per column, but black star(s); Figure 4b). 

For 3 of the 36 measured electrodes, in 3 of the 20 patients, no contact was found in or touching the ventral subpart of the probabilistic volume with β-oscillations and no contact was measured with significantly stronger β-oscillations than those computed over all contacts of the electrode (Figure 3, no yellow-filled cells and no black asterisk per column; Figure 4c). For one electrode (patient P6, right electrode), although the most ventral contact was still in contact with the dorsal part of the probabilistic volume with β-oscillations, no significantly stronger β-oscillations were found. 

In addition, dorsal to the STN, β-oscillations were found on contacts located in or touching the Campus Forelii (fields of Forel), the ZI, or the internal capsule (IC) close to the Campus Forelii or ZI for 11 electrodes (Figure 3, upper line, all but three of the cells with grey asterisks; Appendix A). In three medial electrodes, β-oscillations were found on contacts located in the thalamic ventral-oralis posterior (VLa) or ventral oralis anterior nucleus (Figure 3, upper line, 3 of 14 cells with grey asterisks, Appendix A); patients P3 left, P15 right, P6 left electrodes). For all three of these electrodes, the significance of the measured β-oscillations was low (*p* < 0.01). For deeply inserted electrodes, β-oscillations were found on contacts located in the substantia nigra (Figure 3, lower line, grey asterisks, Appendix A; patients P8 left, P13 left, P1 left electrodes). For two of these three deep electrodes, the significance of the measured β-oscillations was low (*p* < 0.01).

### 3.2. Electrode Location and Motor Outcome

The motor scores were not available for two patients (P1 and P9). Figure 5 indicates the anatomical structures of the stimulated contacts as obtained with the Lead-DBS DISTAL atlas [6] in 18 patients for 33 electrodes. The patients were ordered with decreasing lateralized MDS-UPDRS part III improvements of the contralateral hemibody, as stimulated 1 year post-surgery. Patients were then distributed in a post hoc analysis over three categories, as shown in Figure 5; if at least one stimulated contact was found external, dorsal and/or lateral to the STN (dorsal/lateral: ZI, Campus Forelii, IC), its associated improvement in the MDS-UPDRS score was attributed to the dorsal/lateral group (three right and five left electrodes in five patients). If stimulated electrode contacts were located in the STN, without any contact in the nucleus reticulatus polaris (NRP), the thalamus, or dorsal and/or lateral to the STN, the associated improvements in the MDS-UPDRS score were attributed to the STN category (eight right and seven left electrodes in 11 patients). If at least one stimulated contact was found in the NRP or in a thalamic nucleus, its associated improvement in the MDS-UPDRS score was attributed to the medial group (six right and four left electrodes in seven patients).

Figure 6a illustrates the location of all electrodes for which at least one stimulated contact was found dorsal and/or lateral to the STN, i.e., the dorsal/lateral group; Figure 6b illustrates the location of all electrodes of the STN category, or central group; and Figure 6c illustrates the location of all electrodes of the medial category.

Figure 7 shows the improvements in MDS-UPDRS motor scores 1 year post-surgery for the three groups of patients. Statistically significant differences were found among the groups (Kruskal–Wallis one-way ANOVA on ranks, *H* (2) = 15.2, *p* < 0.001). The largest improvements were seen in patients for whom at least one stimulated contact was found dorsal and/or lateral to the STN (Figure 7, purple circles, eight scores, median 64.1%, percentiles: 25th, 60.9%; 75th, 77.4%). Their scores were significantly higher than those in patients for whom contacts were found within the STN and with no stimulated contacts dorsal and/or lateral to the STN, or in the NRP, or in the thalamus (multiple comparison procedure, Dunn’s method, *p <* 0.05; orange circles, 15 scores: median 39.0%, percentiles: 25th, 33.0%; 75th, 46.0%) and higher than those in patients for whom at least a stimulated contact was found more medially, in the NRP or in a thalamic nucleus (multiple comparison procedure, Dunn’s method, *p* < 0.05; light green scores, 9 scores: median 21.5%, percentiles: 25th, −19.8%; 75th, 40.8%). Note that the right electrode of patient P18 was only stimulated in the zona incerta and was hence not included in the comparison.

Among the group of patients with at least one stimulated contact located dorsal and/or lateral to the STN, patient P6 experienced a slightly husky voice, patients P11 and P4 experienced slight dysarthria, and patient P12 experienced a temporary deviation of the lip towards the right (patients with asterisk, Figure 5). No capsular side effect was observed for the other DBS patients in their daily life.

As described in Figure 2 in the background and in Appendix A (anterior view), the STN-DBS target red volume [7] is located just adjacent and dorsally to the ventral part of the probabilistic volume with β-oscillations. Hence, the present patient series does not confirm the STN-DBS target location as indicated by Lead-DBS. Instead, the study suggests the dorsal and/or lateral external border of the STN, but with the risk of capsular side effects.

Regarding the intraoperative electrophysiological targeting of the STN, at the time of intraoperative stimulation, 9 of 40 electrodes (including electrodes for which β-oscillations were not recorded) were placed on trajectories added to those used for microelectrode recordings, whether or not recordings of the central and/or posterolateral microelectrodes showed typical STN cells. No microelectrode recordings were performed for these additional trajectories in order to avoid having to move the set of exploratory electrodes up and down again. In one of nine cases, a lateral electrode was inserted in addition to the exploratory central and posterolateral electrodes, contributing to the placement of the final DBS electrode lateral to the sensorimotor STN (patient P6, left electrode). In eight of nine cases, a medial or a posteromedial electrode was added in addition to the central and posterolateral trajectories. In six of these eight cases, the added electrode contributed to a deterioration in the targeting of the sensorimotor STN, with the electrode being placed medially in the STN or medially to it (patient P7, right electrode; patient P4, left electrode; patients P13 and P19, both electrodes; Appendix A). On the other hand, the added trajectory contributed to an improvement in the targeting of the sensorimotor STN for two of eight electrodes (patient P5, left electrode; patient P10, left electrode; Appendix A). Thus, overall, six of nine added electrodes contributed to a deterioration in the targeting of the sensorimotor STN. In 3 of the other 31 STNs (patient P1, right electrode; patient P2, right electrode; patient P15, right electrode; Appendix A), the definitive electrode was located too medially, with no contact in the STN, where no trajectory was added at the time of stimulation.

## 4. Discussion

This study, based on high-quality β-oscillation measurements from LFPs of the Vercise Cartesia directional electrode contacts, verifies for the first time the accuracy of the location of STN β-oscillations as indicated by the Lead-DBS MATLAB toolbox. In particular, the results validate the ventral part of the probabilistic banana-like shape in the dorsolateral sensorimotor STN in a different dataset from a different center than that used for the establishment of the atlas. 

In addition, the results are congruent with the existence of a very efficient stimulation area for reducing motor symptoms that is adjacent but external to the dorsolateral sensorimotor subregion of the STN. This was found including all electrode contacts, whatever their location and without any manual adjustment of electrode imaging reconstruction. This stimulation area was more efficient than the area of the actual STN-DBS target for reducing motor symptoms. Nonetheless, lateral to the sensorimotor STN, the stimulation can induce corticospinal tract stimulation (i.e., capsular effects). The robustness of intraoperative stimulation during STN-DBS surgery was not supported by the present study, which instead supports the development of intraoperative imaging reconstruction tools for STN-DBS surgery.

### 4.1. β-Oscillations

The Lead-DBS toolbox reconstructed the location of STN β-oscillations [8], as could be verified with intraoperative measurements. β-oscillations measured on the electrode contacts were confirmed to be located in the ventral part of the dorsolateral sensorimotor STN subdivision. The high agreement found between probabilistic and measured β-oscillations supports the use of the Lead-DBS toolbox to determine their location postoperatively. The ventral part of the banana-like shape probabilistic volume with β-oscillations was the site of maximum measured β-oscillations.

β-oscillations were computed from LFPs, which were developed early in the STN-DBS field [1], but are recorded here in the monopolar rather than bipolar mode. Indeed, using the shielded cables of the Neurostar electrophysiological system, we intraoperatively placed a neutral reference at the U-shaped skin flap, free of brain activity, allowing monopolar recordings while eliminating the intrinsic contamination to the referencing of bipolar measurements. Note that the techniques of monopolar recordings were performed here without common averages nor intracranial referencing or excluding any contacts. The bipolar recordings used to build the electrophysiological atlas could have contributed to the banana-like shape of the β-oscillations in the probabilistic volume [8].

Furthermore, if the difference in the surfaces of non-segmented contacts had been integrated, the spectral density would have to be increased by 15% for the most ventral contacts, 1 and 8. This would have again favored assigning even stronger β-oscillations to the ventral banana-like shape of the probabilistic location of β-oscillations (e.g., patient P6, left STN, contact 1 in addition to or in place of contact 4). If this integration had been carried out, it would have been suspected that it was in order to support the findings of the present study. We have hence chosen to analyze the raw data without inclusion of that surface adaptation.

The β-oscillations found close to the Campus Forelii and close to the VLa could be due to the proximity of tracts involved in the pallido-subthalamic [22], or cortico-subthalamic networks [23] of the exaggerated β-oscillations in patients with PD. 

### 4.2. Electrode Location and Motor Outcomes

The external dorsal and/or lateral borders of the sensorimotor STN do not correspond to the location of the STN-DBS target in the Lead-DBS toolbox. The toolbox suggests that the STN-DBS target would be close to the contacts located within the dorsal part of the banana-like shape of the β-oscillation volume, as illustrated in Figure 2 [7].

The patients in the study were the first to receive the Vercise Cartesia directional electrodes in our center. The possibility that the neurologists in charge of selecting which contacts to stimulate were biased by the motivation to use the segmented contacts cannot be excluded. Whatever the methods applied for choosing the stimulation parameters 1 year post-surgery, the location of the electrodes, spanning from lateral to medial, offered the possibility to analyze the efficacy of stimulating the anatomical structures. Note that all contacts, either located inside or outside the STN, were included in the study, which is not always the case in the literature.

For improvement in rigidity in particular [24], the external dorsal border of the sensorimotor STN may be preferred [25]. The caudal field of Forel—or Campus of Forel/Campus Forelii—or the lenticular fasciculus could be involved in the efficacy of the stimulation of contacts dorsal to the sensorimotor STN. The lateral border of the STN has also been identified as an efficient stimulation site [26,27], as has the activation of the hyperdirect loop [28,29]. Several other tracts are located there, as again recently illustrated [30]. Lateral to the STN is the subthalamic fasciculus or pallidosubthalamic bundle, which “terminates in the lateral part of the STN”, as illustrated with 11.7 Tesla MRIs [31]. The subthalamic fasciculus might also be an efficient site for motor improvement, in agreement with the excellent efficacy of globus pallidus externa stimulation [9]. Nevertheless, the lateral border of the STN can induce capsular effects from corticospinal tract stimulation that can prevent the use of contacts. Access to segmented contacts, knowing its precise location, should become an advantage in this context. On the other hand, trajectories going through the NRP or from the VLa of the thalamus, i.e., more medial trajectories, did not result in efficient DBS targets for PD [32].

Concerning the fact that the beta-oscillations are a good marker of the success of STN-DBS [12,33,34,35], if the electrode array crosses the site of beta-oscillations in the STN, the case will indeed be successful. That said, an electrode contact just dorsally or laterally adjacent to the probabilistic volume of beta-oscillations may even produce a stronger benefit than the stimulation of the electrode contact directly placed in the probabilistic volume of beta-oscillations in the sensorimotor STN (Figure 5 and Figure 7). In all three cases, the stimulated electrode contact located at the site of probabilistic beta-oscillations, or adjacent to it dorsally or laterally, will be a success, which is not the case for contacts located medially to the probabilistic volume of beta-oscillations.

### 4.3. Considerations for DBS Surgery

In other respects, the study suggests that intraoperative evaluation of corticospinal tract excitation failed to improve the efficacy of DBS [12,36]. The effects of corticospinal tract excitation can be confused with dyskinesia secondary to STN stimulation. Indeed, among the nine electrodes placed on added trajectories at the time of surgery based on intraoperative stimulation, only one contributed to getting closer to the lateral STN border; the others led to being closer to the medial STN border, which resulted in weak motor improvements. Note that because of the Ben-Gun array geometry with a distance of 2.8 mm between the posteromedial and posterolateral trajectories, and considering the size and shape of the STN, there was a very poor chance that an added medial trajectory could go through the STN when microelectrode recordings indicated STN cells on the central and posterolateral trajectories, for instance. Instead of intraoperative stimulation, the posterolateral STN could be targeted through the measurement of β-oscillations originating in the sensorimotor STN of the macro-contact of the explorative microelectrode in conjunction with microelectrode recordings. 

As the Lead-DBS toolbox indicates contacts with β-oscillations, an imaging reconstruction suite as precise as it is, and its implemented anatomical atlases, should ideally be considered for intraoperative use with the acquired intraoperative 3D images.

## 5. Conclusions

This study, based on β-oscillation measurements from monopolar LFPs of the Vercise Cartesia directional electrode contacts, verifies the accuracy of the location of β-oscillations, as indicated by the Lead-DBS MATLAB toolbox. In particular, the results validate the ventral part of the probabilistic location in the dorsolateral sensorimotor STN. Moreover, the largest improvements of motor symptoms were found with DBS electrodes located on the dorsal and/or lateral external borders of the sensorimotor STN. The robustness of intraoperative stimulation during STN-DBS surgery was not supported by the present study, which instead promotes the development of intraoperative imaging reconstruction tools for STN-DBS surgery.

## Figures and Tables

**Figure 1 bioengineering-10-00898-f001:**
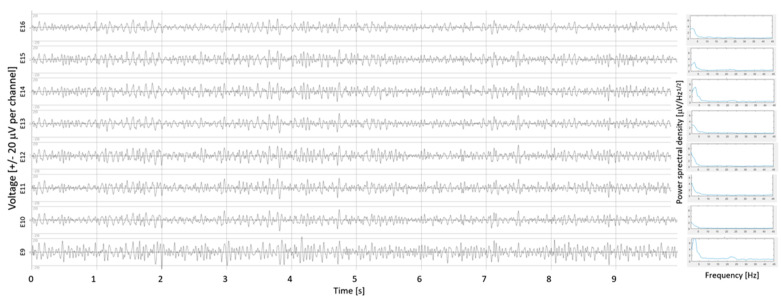
Example of local field potentials (LFPs) recorded intraoperatively on electrode contacts in patient P4 (right electrode), with the power spectral density (PSD) of each contact on the right. Intraoperative LFPs recorded on all eight contacts of the electrodes (band pass filtering (13–31 Hz)). E9, the most ventral contact on the right, bottom trace; E16, the most dorsal contact on the right, top trace; directional contacts, in between. For this electrode, stronger β-oscillations were measured on contact E9 (+130%, *p* < 1.10^−12^); on the right, the PSDs (μV/Hz^1/2^) are displayed for each contact.

**Figure 2 bioengineering-10-00898-f002:**
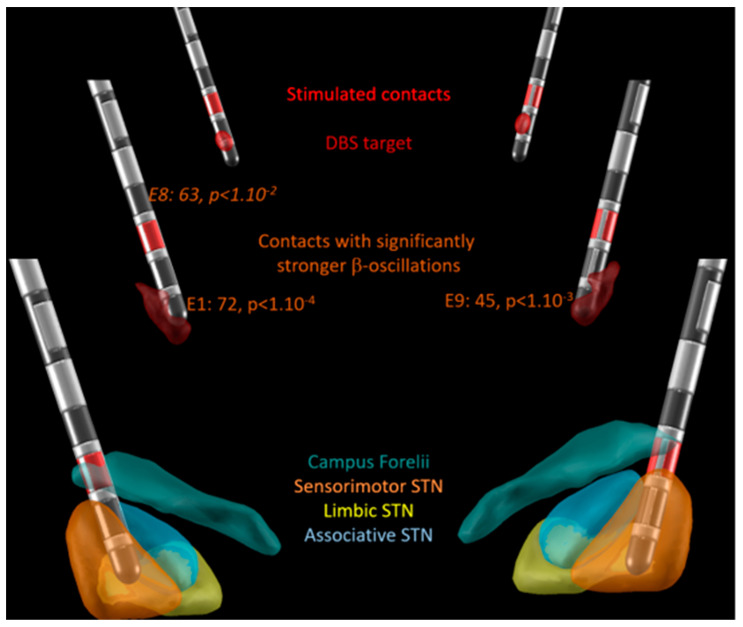
Postoperative image reconstruction of the electrodes performed with Lead-DBS in patient P16 (posterior view). Foreground: Reconstructed location of the subthalamic nucleus sensorimotor (orange), limbic (yellow), and associative (blue) subregions; Campus Forelii (grey green, DISTAL atlas), with stimulated contacts shown in red on all three planes. Second plane: Probabilistic location of β-oscillations (burgundy volume) with statistics of measured β-oscillations. If significantly stronger, the percentage of the difference in power of the β-oscillations of a contact, in comparison to the power of the β-oscillations of all eight electrode contacts, is indicated with the significance of the difference (E1, left most ventral contact; E9, right most ventral contact; E8, left most dorsal contact; directional contacts, in between). Background: STN-DBS probabilistic stimulation target according to the DBS target atlas (red volume in background).

**Figure 3 bioengineering-10-00898-f003:**
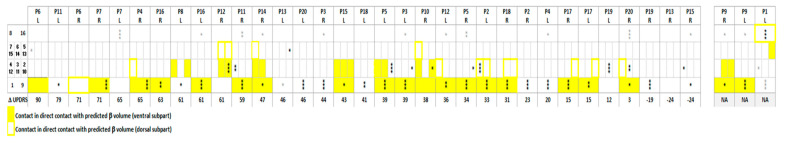
Graphical representation of contacts found in, or touching, the probabilistic volume with β-oscillations by Lead-DBS. Filled yellow boxes are contacts in or touching the ventral subpart of the probabilistic volume with β-oscillations; unfilled yellow boxes are contacts in or touching the dorsal subpart of the probabilistic volume with β-oscillations as computed by the Lead-DBS toolbox. Contacts with measured stronger β-oscillations are indicated by asterisks (* *p* < 0.01; ** *p* < 0.001, *** *p* < 0.0001; grey asterisks for contacts found outside the subthalamic nucleus, refer to Appendix A). For all patients (P, in column), for left (L) and right (R) electrodes, the contacts of the electrodes are represented in rows: contacts E1 (**left**) and E9 (**right**), the most ventral contacts; contacts E8 (**left**) and E16 (**right**), the most dorsal contacts; directional contacts, smaller boxes in between. Electrodes are ordered with decreasing improvements in lateralized MDS-UPDRS part III scores of the contralateral hemibody, as stimulated 1 year post-surgery (NA, not available for two patients).

**Figure 4 bioengineering-10-00898-f004:**
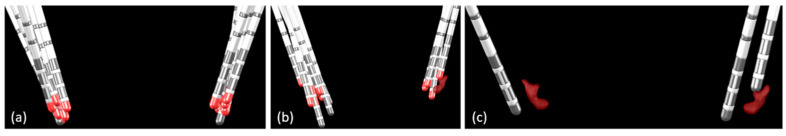
Postoperative image reconstruction of the electrodes, with the probabilistic β-oscillations performed with Lead-DBS (posterior view). Contacts measured with significantly stronger β-oscillations than those computed over all electrode contacts (red contacts) were found for either (**a**) electrodes with at least one contact in or touching the ventral subpart of the probabilistic volume with β-oscillations, or (**b**) electrodes not in or touching the probabilistic volume with β-oscillations. (**c**): electrodes not found with β-oscillations nor adjacent to the probabilistic ventral part of the β-oscillations.

**Figure 5 bioengineering-10-00898-f005:**
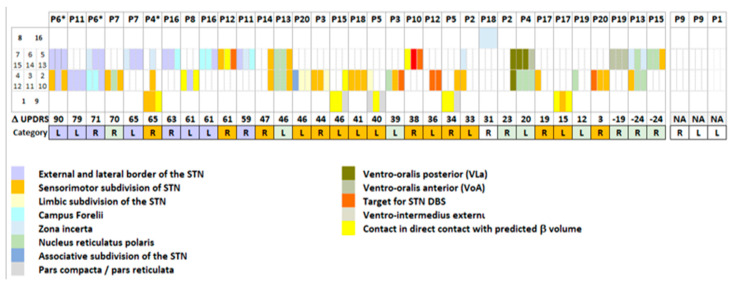
Graphical representation of the anatomical structures where stimulated contacts were located. Anatomical structures were found by applying the Lead-DBS DISTAL atlas. The stimulated contacts are those stimulated 1 year post-surgery at the time of the evaluation of the lateralized MDS-UPDRS part III scores, reported in the last row of the matrix. For all patients (P), for left (L) and right (R) electrodes, the contacts of the electrodes are represented in rows: contacts 1 (left) and 9 (right), the most ventral contacts; contacts 8 (left) and 16 (right), the most dorsal contacts; directional contacts, in between. Electrodes are ordered with decreasing motor improvements (penultimate row; NA, not available for two patients). The color categories represented in the bottom row are the following: purple: at least one stimulated contact in the external dorsal/lateral borders of the STN (Campus Forelii, CF, or internal capsule, IC); orange: stimulated contacts in the STN without any stimulated contact in the CF or IC, nucleus reticulatus polaris (NRP) or thalamus; green: patients with at least one stimulated electrode contact in the NRP or in the thalamus. * Patients with capsular side effects.

**Figure 6 bioengineering-10-00898-f006:**
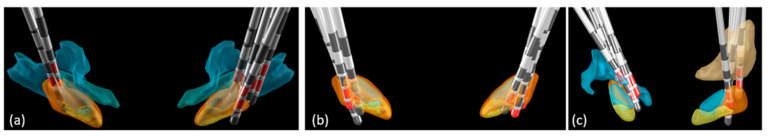
Postoperative image reconstruction of the electrodes (anterior view) with the stimulated contacts (red contacts). Anatomical structures were found by applying the Lead-DBS DISAL atlas. The stimulated contacts are those stimulated 1 year post-surgery at the time of the evaluation of the lateralized MDS-UPDRS part III scores: (**a**) for electrodes with at least one stimulated contact in the Campus Forelii (CF) or the internal capsule (IC); (**b**) for electrodes with stimulated contacts in the subthalamic nucleus without any contact in the CF or in the IC, in the nucleus reticulatus polaris (NRP), or in the thalamus; (**c**) for electrodes with at least one stimulated electrode contact in the NRP or in the thalamus.

**Figure 7 bioengineering-10-00898-f007:**
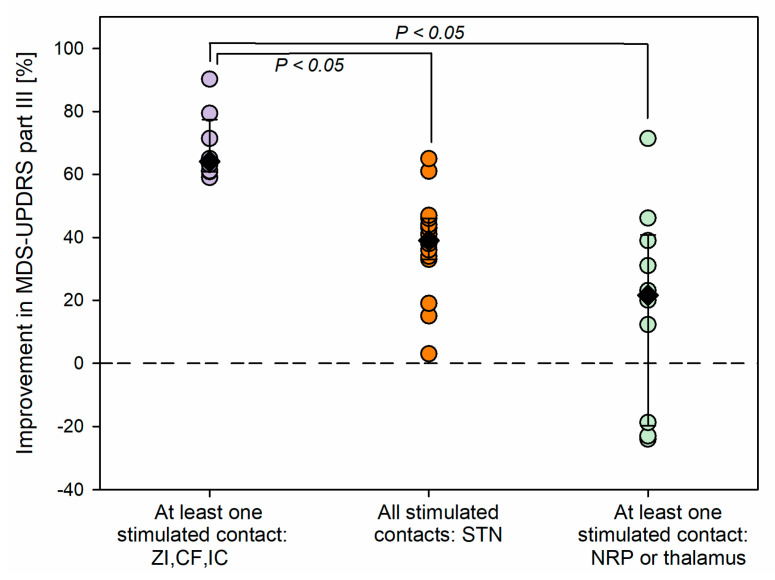
Comparison in improvements in the MDS-UPDRS part III scores of the contralateral hemibody, as stimulated 1 year post-surgery for different anatomical structures. The color categories representing different configurations of stimulated contacts are the same as the ones in Figure 5.

## Data Availability

Apart from the open-access Lead-DBS toolbox (www.lead-dbs.org (accessed on 21 January 2022)), the MATLAB code used for analyzing β-oscillations will be shared upon request to C.B. The raw data will be shared, respecting the limitations of the University Hospitals of Geneva and in agreement with our local Ethics Committee, through the University of Geneva (http://www.kheops.ch/ (accessed on 19 October 2022)).

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
