# Peer review of "Validation of Lead-DBS β-Oscillation Localization with Directional Electrodes"

_bioengineering, 2023, doi:10.3390/bioengineering10080898_

Round 1

Reviewer 1 Report

The authors presented an interesting study about validating the beta oscillation localization implemented in Lead-DBS using intraoperative local field potential recordings from 20 patients (36 leads). The results confirmed the ventral subpart of the probabilistic beta-oscillations but did not confirm the STN-DBS target location as indicated by Lead-DBS, which was a bit unexpected but also interesting. I have a few comments hopefully can be useful for improvement.

1. As the results confirmed the beta oscillations localization but not the STN-DBS target location as indicated by Lead-DBS, which somehow challenged previous studies in which beta was suggested to be a good biomarker of DBS effect in PD (Neumann et al., 2016; Kühn et al., 2006; Kehnemouyi et al., 2021). Could you please comment on this? It would also be interesting to further check that within this cohort, if the reduction of UPDRS-III with DBS correlated with either beta reduction or baseline beta on the stimulating contact.

Neumann, W.J., Degen, K., Schneider, G.H., Brücke, C., Huebl, J., Brown, P. and Kühn, A.A., 2016. Subthalamic synchronized oscillatory activity correlates with motor impairment in patients with Parkinson's disease. Movement Disorders31(11), pp.1748-1751.

Kühn, A.A., Kupsch, A., Schneider, G.H. and Brown, P., 2006. Reduction in subthalamic 8–35 Hz oscillatory activity correlates with clinical improvement in Parkinson's disease. European Journal of Neuroscience23(7), pp.1956-1960.

Kehnemouyi, Y.M., Wilkins, K.B., Anidi, C.M., Anderson, R.W., Afzal, M.F. and Bronte-Stewart, H.M., 2021. Modulation of beta bursts in subthalamic sensorimotor circuits predicts improvement in bradykinesia. Brain144(2), pp.473-486.

2. Method: The statistic applied to test the significance of beta-oscillations on each individual contact against all contacts was not clear. How exactly did you apply student t-test between value(s) for one contact and values for all 8 contacts (within the electrode)?

3. Figure 1 is not very informative, as it is difficult to see the difference in beta just from filtered time-series. Maybe add PSD plots for each contact and illustrate the method that used for statistical analysis.

4. Figure 2 is very confusing. Please either use more distinguishable colours or add labels on the figure.

5. Figure 3 is very difficult to read. Please consider present the results in another way.

6. Figure 5: As it is, the three categories are not identified from the figure. Please consider adding notes/marks to better group the hemispheres into those categories.

Author Response

Responses to Reviewer 1:

“The authors presented an interesting study about validating the beta oscillation localization implemented in Lead-DBS using intraoperative local field potential recordings from 20 patients (36 leads). The results confirmed the ventral subpart of the probabilistic beta-oscillations but did not confirm the STN-DBS target location as indicated by Lead-DBS, which was a bit unexpected but also interesting. I have a few comments hopefully can be useful for improvement.”

“1. As the results confirmed the beta oscillations localization but not the STN-DBS target location as indicated by Lead-DBS, which somehow challenged previous studies in which beta was suggested to be a good biomarker of DBS effect in PD (Neumann et al., 2016; Kühn et al., 2006; Kehnemouyi et al., 2021). Could you please comment on this? It would also be interesting to further check that within this cohort, if the reduction of UPDRS-III with DBS correlated with either beta reduction or baseline beta on the stimulating contact.”

Thank you for this interesting and important point, that gives the opportunity to better discuss this point.
We agree with the fact that the beta-oscillations are a good marker of the success of STN-DBS. Actually, we have also shown that the beta-oscillations were the strongest marker of the success of STN-DBS (Boëx, et al., “What is the best electrophysiologic marker of the outcome of subthalamic nucleus stimulation in Parkinson disease?, 2018). Indeed, the fact that the electrode array crosses the site of beta-oscillations in the STN is predictive of a successful case. That said, an electrode contact placed just dorsally or laterally adjacent to the probabilistic volume of beta-oscillations may even produce a stronger benefit than the stimulation of the electrode contact directly placed into the probabilistic volume of beta-oscillations in the sensorimotor STN (Figure 5 and 7). In all three cases, the stimulated electrode contact located at the site of maximum beta-oscillations, or adjacent to it dorsally or laterally, will all be a success, which is not the case for contacts located medially adjacent to the probabilistic volume of beta-oscillations. We added a few sentences in the discussion about this point.

That said, we did not have access to the electrode outside of the operative room and could not measure the effect of the stimulation on the beta-oscillations. We added these references in the introduction concerning the modulation of beta-oscillations.

«Neumann, W.J., Degen, K., Schneider, G.H., Brücke, C., Huebl, J., Brown, P. and Kühn, A.A., 2016. Subthalamic synchronized oscillatory activity correlates with motor impairment in patients with Parkinson's disease. Movement Disorders, 31(11), pp.1748-1751.”
Kühn, A.A., Kupsch, A., Schneider, G.H. and Brown, P., 2006. Reduction in subthalamic 8–35 Hz oscillatory activity correlates with clinical improvement in Parkinson's disease. European Journal of Neuroscience, 23(7), pp.1956-1960.
Kehnemouyi, Y.M., Wilkins, K.B., Anidi, C.M., Anderson, R.W., Afzal, M.F. and Bronte-Stewart, H.M., 2021. Modulation of beta bursts in subthalamic sensorimotor circuits predicts improvement in bradykinesia. Brain, 144(2), pp.473-486.”
Referecences added

“2. Method: The statistic applied to test the significance of beta-oscillations on each individual contact against all contacts was not clear. How exactly did you apply student t-test between value(s) for one contact and values for all 8 contacts (within the electrode)?”
We added further precision, for making clear that the PSDs (1s, 1Hz) of one contact were compared to all PSDs of all eight contacts, making the comparison even more severe.

“3. Figure 1 is not very informative, as it is difficult to see the difference in beta just from filtered time-series. Maybe add PSD plots for each contact and illustrate the method that used for statistical analysis.”
We added the PSDs of each contact in addition to the raw LFP that we still find informative.

“4. Figure 2 is very confusing. Please either use more distinguishable colours or add labels on the figure.”
Because the colors are from the Lead-DBS software, we cannot modify them. We therefore added labels to Figure 2.

“5. Figure 3 is very difficult to read. Please consider present the results in another way.”
Finding one way to group the results of all electrode contacts of all patients has been a challenge. We could not imagine another way to do it after having tested other solutions that were not found to be better. We still would like to keep the solution of an array. We improved the fonts in this updated Figure 3 and increased its size. More precision was also added to the text to help in the interpretation of Figure 3.

“6. Figure 5: As it is, the three categories are not identified from the figure. Please consider adding notes/marks to better group the hemispheres into those categories.”
This point helps make the group data clearer, thank you. As for Figure 3, we improved the fonts, increased the size, and hence categorized each electrode into the categories used in Figure 7. We also loved the hemisphere side in this new line.

Reviewer 2 Report

bioengineering-2490191: “Validation of Lead-DBS β-oscillation localization with directional electrodes

In this study, the authors are trying to demonstrate that in PD patients, the lead-DBS toolbox reconstruction of β-oscillations in the subthalamic nucleus coincides with their probabilistic intraoperative location evaluated with “directional electrodes”. The efficacy of DBS stimulation of the subthalamic area in the motor symptoms improvement is analyzed as well.

The material is described successively and conclusions are partially supported by obtained data.

Remarks/recommendations:

  1. the title suggestion: “Validation of Lead-DBS-associated LFP β-Oscillations with Directional Electrodes localized in the Subthalamic Nucleus”;
  2. in lines 16 and 23, abbreviations of “Lead-DBS” and “DISTAL” should be open;
  3. in lines 26-30, the sentence should be simplified and grammatically corrected;
  4. in the section of “2.1. Patients and Clinical Assessment”, the ethical guidelines used and the ethical committee approvement should be clarified;
  5. in line 74, “consecutive” might be removed;
  6. Lead-DBS parameters should be mentioned;
  7. in line 96, “GE” should be replaced by “Germany”;
  8. in line 132, “…[16] was…”;
  9. in line 138, “…the DiODe technic [21].”
  10. in lines 152,153, it is unclear what “Power values (309; [7-35 Hz])” means and how many patients were used in this study (see line 74);
  11. Figures 1 and 3 needs more visible fonts;
  12. in line 202, for “electrodes”, the unified formatting should be used;
  13. in lines 379-383 and 398-402, the sentences should be simplified;
  14. in line 481, “…was not supported…”

English should be double-checked.

Moderate editing of English language required

Author Response

bioengineering-2490191: “Validation of Lead-DBS β-oscillation localization with directional electrodes”

“In this study, the authors are trying to demonstrate that in PD patients, the lead-DBS toolbox reconstruction of β-oscillations in the subthalamic nucleus coincides with their probabilistic intraoperative location evaluated with “directional electrodes”. The efficacy of DBS stimulation of the subthalamic area in the motor symptoms improvement is analyzed as well.

The material is described successively and conclusions are partially supported by obtained data.”

Remarks/recommendations:

  1. “the title suggestion: “Validation of Lead-DBS-associated LFP β-Oscillations with Directional Electrodes localized in the Subthalamic Nucleus”;”

Because all electrode contacts were not always in the STN, and that we would like to avoid too many acronyms in the tittle, not sure how we could change the title.

  1. “in lines 16 and 23, abbreviations of “Lead-DBS” and “DISTAL” should be open;”

We have explained the DISTAL acronym in line 23. However, “Lead-DBS” is the name of the software and does not have an “open” version. DBS obviously stands for Deep Brain Stimulation, and this acronym is defined further.

  1. “in lines 26-30, the sentence should be simplified and grammatically corrected;”

Thank you for your remark, the sentence has been corrected and simplified.

  1. “in the section of “2.1. Patients and Clinical Assessment”, the ethical guidelines used and the ethical committee approvement should be clarified;”
    It has been added
  1. “in line 74, “consecutive” might be removed”;

Done

  1. “Lead-DBS parameters should be mentioned;”

The paragraph 2.3. Postoperative Image reconstruction specifies all parameters applied.

  1. “in line 96, “GE” should be replaced by “Germany”;”

Done

  1. “in line 132, “…[16] was…”;

Done

  1. “in line 138, “…the DiODe technic [21].”

Done

  1. “in lines 152,153, it is unclear what “Power values (309; [7-35 Hz])” means and how many patients were used in this study (see line 74);”

“309” seems just to have no meanings at all ; seems a misprint; there’s no link with the number of patients included as described in line 74

  1. Figures 1 and 3 needs more visible fonts;”

We improved the visibility of Figures 1, 3 and 5, including the changes in fonts

  1. “in line 202, for “electrodes”, the unified formatting should be used;”

Done

  1. “in lines 379-383 and 398-402, the sentences should be simplified;”

Done

  1. “in line 481, “…was not supported…”

Done

“English should be double-checked.”

The manuscript has been reviewed by a professional English editor. We made it controlled again and simplified the requested sentences.

Round 2

Reviewer 2 Report

bioengineering-2490191: “Validation of Lead-DBS β-oscillation localization with directional

electrodes”

The authors have made a careful revision and reasonably responded to almost all points I raised.

Suggestions and remarks:

Abstract: In deep brain stimulation (DBS) studies in patients with Parkinson’s disease, the Lead-DBS toolbox allows the reconstruction of the location of β-oscillations in the subthalamic nucleus (STN) by using of Vercise Cartesia directional electrodes (Boston Scientific). The objective was to compare these probabilistic locations with those of intraoperative monopolar β-oscillations computed from local field potentials [0.5-3kHz] recorded by using shielded single wires and an extracranial shielded reference electrode.

In lines 22,23: “…of all eight contacts on the directional electrodes.”  

In lines 25,26: “…16/20 both, 4/20 single STN)….” needs to be clarify.

In line 27: “In 33 of 36 electrodes…., 10 electrodes did not touch it but were adjacent to this volume; in 3 of 36 electrodes,…”.

Figure 1:    1) the brain areas should be denoted by more visible fonts;

                   2) the amplitude calibration should be either denoted by enlarged fonts on the plate or mentioned in the legend as ±20 µV.

Minor editing of English language required

Author Response

bioengineering-2490191: “Validation of Lead-DBS β-oscillation localization with directional electrodes”

“The authors have made a careful revision and reasonably responded to almost all points I raised.

Suggestions and remarks:

Abstract: In deep brain stimulation (DBS) studies in patients with Parkinson’s disease, the Lead-DBS toolbox allows the reconstruction of the location of β-oscillations in the subthalamic nucleus (STN) by using of Vercise Cartesia directional electrodes (Boston Scientific). The objective was to compare these probabilistic locations with those of intraoperative monopolar β-oscillations computed from local field potentials [0.5-3kHz] recorded by using shielded single wires and an extracranial shielded reference electrode.”

Thank you very much. We changed the abstract to modify this paragraph.

In lines 22,23: “…of all eight contacts on the directional electrodes.”  
Done

In lines 25,26: “…16/20 both, 4/20 single STN)….” needs to be clarify.

We added “electrodes” to STN, replaced “/” by “of”.

In line 27: “In 33 of 36 electrodes…., 10 electrodes did not touch it but were adjacent to this volume; in 3 of 36 electrodes,…”.

 We changed the sentence.

 Figure 1:    1) the brain areas should be denoted by more visible fonts;

                   2) the amplitude calibration should be either denoted by enlarged fonts on the plate or mentioned in the legend as ±20 µV.

Oh yes the new figure was not downloaded, here it is, thank you.